# Mine Productivity Upper Bounds and Truck Dispatch Rules

Adriano Chaves Lisboa [1,*,†] , Felipe Luz Barbosa Castro [2] and Pedro Vinícius Almeida Borges de Venâncio [1]

1   Gaia, Solutions on Demand, Belo Horizonte 31310-260, Brazil; pedro.venancio@gaiasd.com
2   Vale S.A., Nova Lima 34006-049, Brazil; felipe.castro@vale.com
*   Correspondence: adriano.lisboa@gaiasd.com
†   Current address: José Vieira de Mendonça 770, Belo Horizonte 31310-260, Brazil.

**Abstract:** This paper proposes an upper bound for mine productivity (useful for long-term planning) and also a simple truck dispatch rule (useful for short-term operations) that demonstrates how tight the upper bound can be using a simulation. It also proposes a greedy search to approximate the productivity upper bound, which is faster and often exact. Uncertainty is added to the simulation in order to verify how the productivity responds to it. Typically, the productivity's upper bound is less tight close to its saturation point as a function of the number of trucks, where adding more trucks only increases queues. Furthermore, more uncertainty in the model typically leads to a less tight upper bound. The results conducted using real data from an open pit mine in Brazil show that the gap between the productivity upper bound and the productivity realization using the proposed dispatch rule for a homogeneous fleet can be less than 2%, but it can be as large as 12% near the productivity saturation point without uncertainty. Even though this gap seems to become arbitrarily small as the number of trucks and the simulation horizon increase, the productivity upper bound is never violated, which validates it as an upper bound and induces optimality for the dispatch rule.

**Keywords:** mine productivity; upper bound; truck dispatch rule





## 1. Introduction

Mining is a process of extracting valuable materials from underground and open pit mines. These materials, known as ores, are typically a combination of minerals, natural rocks, and sediments, which have economic value when refined. During the last few decades, mining has played an important role in the economic development of several countries, especially in emerging countries [1]. However, mining is a very complex industrial operation, and its progress must contain several planning stages, starting with prospecting for ore bodies and ending with the final reclamation of the land after the mine is closed. In addition, a mining project must maximize the net present value (NPV), extracting the ore at the lowest possible cost over the mine's life cycle and, therefore, making the effort of such labor worthwhile.

In open pit mines, where operating costs are very high, it is even more essential to maximize the productivity of a minimum cost. Among the most expensive open pit mine operations, haulage and material handling stand out, accounting for 50–60% of the total operating costs [2]. In order to reduce this cost and thereby maximize the NPV of the mining project, a fleet management system (FMS) is required, whose goal is to solve two problems: (i) find the shortest path to travel between each pair of locations (loading and dumping sites) in the mine (shortest path problem) and (ii) determine the number of truck trips required for each path and then dispatch the trucks to the locations in real-time, which is the focus of this paper.

Hence, the FMS can be single-stage or multi-stage. The single-stage approaches dispatch trucks without considering any production targets or constraints and typically consist of heuristics based on rules of thumb [3,4]. However, multi-stage approaches have a

great advantage over single-stage approaches by dividing problem into two sequential sub-problems [5,6]. This division into stages introduces the second level of knowledge to the FMS, which improves the quality of the solutions, as well as allowing them to better adapt to real scenarios with uncertainties. The first sub-problem consists of efficiently allocating haulage resources for excavation activities based on truck loads, aiming to maximize the truck productivity (truck and shovel allocation problem—upper stage), and the second sub-problem consists of dispatching trucks to a loading or a dumping site (truck dispatching problem—lower stage) [7].

Although it has not been extensively investigated as the upper stage sub-problem, the truck dispatching problem is essential for the FMS. It is by solving the lower stage sub-problem that planning comes into play to achieve the production targets defined in the previous stage. Formally, the truck dispatching problem, which can be treated as an assignment problem [5,8] or a transportation problem [9–11] is real-time decision making associated with the destinations of trucks to satisfy the production requirements in a mining operation. To achieve these requirements, one or more objectives are usually defined, including the maximization of mine productivity or the minimization of truck inactivity (whether through idle time, waiting time, or loading/dumping time). Therefore, several formulations of this optimization problem have already been proposed, as well as different solutions to these formulations.

Since the truck dispatch problem also occurs in fuel and package deliveries, taxis and ride-hailing services, and other industries that have to manage fleets of vehicles, some approaches used to solve the dispatch problem in other contexts have been naturally adopted for the problem applied in the mining industry. However, the transition of approaches in different contexts can be inappropriate. In open pit mines, some important particularities must be considered in the optimization problem. For example, the travel distance between two locations is usually short, the time taken for the truck to load or dump is often longer than the travel time, and the frequency of demand at each location is often higher [2].

Therefore, the efficiency of a solution to the truck dispatch problem aimed at maximizing the mine's productivity is strictly related to the fleet size and the haulage distances. A fleet with an insufficient number of trucks (under-truck) will result in substantially unproductive periods and a fleet with a high number of trucks (over-truck) may lead to queues for loading or dumping. Thus, several methods have been proposed to select the optimal size of the truck fleet in the truck dispatching problem, i.e., the number of trucks is considered to be a decision variable, which avoids the aforementioned problems. Generally, these methods are based on match factor [12–15], artificial intelligence [16–18], operations research [19–22], life cycle cost analysis [23,24], or discrete event simulation [25–27]. However, the major drawback of these works is that they were developed to address only the equipment selection and sizing problem, in particular, the size of the haulage fleet handling the dump materials, and typically disregard the truck dispatch rules [15]. On the other hand, dispatching rules have been studied apart from fleet sizing, based on optimization models with operations research [28,29], dynamic truck allocation [30,31], heuristics with real-time data [32,33], simulations [34], or artificial intelligence [35]. This paper sheds some light on the interface between long-term fleet sizing with productivity estimation and its realization in the short term with a specific dispatch rule.

From this perspective, we define a linear programming model that derives the upper bound for mine productivity by considering trucks of a heterogeneous fleet allocated in cycles (pairs of loading–dumping sites). In addition, we propose a simple truck dispatch rule that leads to mine productivity, found in a discrete event simulation, close to this upper bound. Through a case study in an open pit mine in Brazil, we also show that the simulation takes into account fleet size problems, and the results obtained can be feasibly implemented in real-world situations.

## 2. Mine Model

The mine model used in this paper considers a productive cycle of individual trucks in a mine which are assigned to a loading site, dispatched to the chosen loading site where they are loaded, assigned to a dumping site, dispatched to the chosen dumping site where they are finally dumped and start the cycle all over again, as shown in Figure 1. The degree of freedom lies in determining where to assign the trucks to load or dump at each moment, which must maximize the mine productivity in the long term. In the next section, an upper bound for mine productivity is proposed using linear programming, which is latter approximated by a greedy search. The mine productivity is physically realized by a discrete event simulation using a simple truck dispatch rule. The loading and dumping points are shared truck resources with respective service times, which may result in truck queues. Truck queues are explicitly modeled in the simulation, where they are used to schedule new events, and they are implicitly modeled during optimization, where trucks are allocated up to the total occupation of the loading or dumping points (i.e., from this point on, more trucks would result only in queues and not in productivity gains).

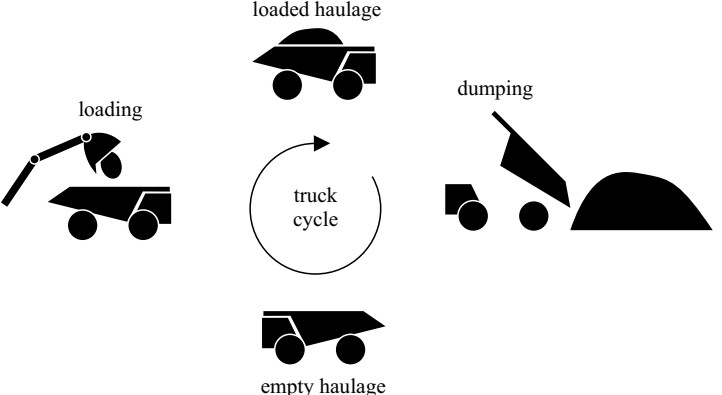

**Figure 1.** Truck cycle with its four states.

### 2.1. Optimization Problem

In order to derive an upper bound for long-term mine productivity, trucks are considered to be allocated in cycles defined by pairs of loading–dumping sites, where the cycle times are given by

$$t_{c,i_u,i_\ell,i_m} = 2\frac{d_{i_u,i_\ell}}{v_{i_m}} + t_{u,i_u,i_m} + t_{\ell,i_\ell,i_m}, \quad \forall i_u, i_\ell, i_m \tag{1}$$

where $d_{i_u,i_\ell}$ is the distance between the dumping site $i_u$ and the loading site $i_\ell$, $v_{i_m}$ is the truck speed for model $i_m$, $t_{u,i_u,i_m}$ is the dumping time at the dumping site $i_u$ for truck model $i_m$, and $t_{\ell,i_\ell,i_m}$ is the loading time at the loading site $i_\ell$ for truck model $i_m$. The trucks are allowed to change their loading and dumping sites after each load or dump operation. Hence, in order to derive an upper bound for the productivity, the number of trucks of model $i_m$ allocated in the cycle defined by loading site $i_\ell$ and dumping site $i_u$ is a non-negative real number $N_{i_u,i_\ell,i_m}$, where fractions denote the truck's relative time slice in each cycle. The resulting productivity of this allocation is given by

$$\frac{N_{i_u,i_\ell,i_m}L_{i_m}}{t_{c,i_u,i_\ell,i_m}} \tag{2}$$

where $L_{i_m}$ is the load of truck model $i_m$, and $t_{c,i_u,i_\ell,i_m}$ is the respective cycle time.

The maximum number of trucks that a resource (e.g., loading or dumping site) can support in a cycle without queues is given by

$$\frac{t_{c,i_u,i_\ell,i_m}}{t_{u,i_u,i_m}} \text{ or } \frac{t_{c,i_u,i_\ell,i_m}}{t_{\ell,i_\ell,i_m}} \tag{3}$$

so that each truck occupies a cycle time slice with the length given by the service time and serviced in a continuous way without queuing. Hence, the occupation due to an allocated number of trucks $N_{i_u,i_\ell,i_m}$ is given by

$$\frac{N_{i_u,i_\ell,i_m} t_{u,i_u,i_m}}{t_{c,i_u,i_\ell,i_m}} \text{ or } \frac{N_{i_u,i_\ell,i_m} t_{\ell,i_\ell,i_m}}{t_{c,i_u,i_\ell,i_m}} \tag{4}$$

which must be, at most, 1 to prevent the generation of queues.

The linear optimization problem for maximum productivity can be written as

$$\text{maximize} \quad \sum_{i_u=1}^{n_u} \sum_{i_\ell=1}^{n_\ell} \sum_{i_m=1}^{n_m} \frac{N_{i_u,i_\ell,i_m} L_{i_m}}{t_{c,i_u,i_\ell,i_m}} \tag{5}$$

$$\text{subject to} \quad \sum_{i_\ell=1}^{n_\ell} \sum_{i_m=1}^{n_m} \frac{N_{i_u,i_\ell,i_m} t_{u,i_u,i_m}}{t_{c,i_u,i_\ell,i_m}} \le 1, \quad \forall i_u \tag{6}$$

$$\sum_{i_u=1}^{n_u} \sum_{i_m=1}^{n_m} \frac{N_{i_u,i_\ell,i_m} t_{\ell,i_\ell,i_m}}{t_{c,i_u,i_\ell,i_m}} \le 1, \quad \forall i_\ell \tag{7}$$

$$\sum_{i_u=1}^{n_u} \sum_{i_\ell=1}^{n_\ell} N_{i_u,i_\ell,i_m} \le n_{t,i_m}, \quad \forall i_m \tag{8}$$

$$0 \le N_{i_u,i_\ell,i_m} \le n_{t,i_m}, \quad \forall i_u, i_\ell, i_m \tag{9}$$

where $N_{i_u,i_\ell,i_m}$ (design variable) is the number of trucks of model $i_m$ allocated in the cycle between the dumping site $i_u$ and the loading site $i_\ell$, $n_{t,i_m}$ is the number of available trucks of model $i_m$, $n_u$ is the number of dumping sites, $n_\ell$ is the number of loading sites, and $n_m$ is the number of truck models. The problem is basically a productivity maximization (5) problem subject to resource constraints (6)–(8).

Greedy Search

The linear problem (5)–(9) maximizes a conical combination (i.e., a linear function with positive coefficients) on the design variables $N$ (number of allocated trucks) subject to upper bounded conical combinations on $N$. If only one resource constraint is considered, the exact solution would be to allocate trucks with the largest benefit–cost ratios, where benefit is be given by the objective coefficients and cost is caused by the constraint coefficients. Considering all three resource constraints, a greedy search is then proposed to find an approximate solution to the linear problem (5)–(9) by allocating trucks to the most productive cycles (i.e., the ones relative to the largest objective coefficients), up to their resource availability bound given by the most restrictive resource constraint.

Algorithm 1 depicts this greedy search. Lines 1 and 2 initialize the output parameters. Lines 3 and 4 initialize the cycle time and cycle productivity. Lines 5–7 initialize the resource (dumping sites, loading sites and trucks) allocation. Line 8 initializes the remaining cycle indicator. Line 10 finds the most productive remaining cycle. Line 11 allocates trucks according to the resource availability. Lines 12–14 update the resource availability. Lines 15 and 16 update the output parameters. Lines 17–26 update the remaining cycle indicator.

---

**Algorithm 1** Greedy search for mine productivity.

---

**Input**

$\quad d \in \mathbb{R}^{n_u \times n_\ell}$ distance between the dumping and loading sites

$\quad t_u \in \mathbb{R}^{n_u \times n_m}$ truck dumping time

$\quad t_\ell \in \mathbb{R}^{n_\ell \times n_m}$ truck loading time

$\quad n_t \in \mathbb{R}^{n_m}$ number of trucks

$\quad v \in \mathbb{R}^{n_m}$ truck speed

$\quad L \in \mathbb{R}^{n_m}$ truck load

**Output**

$\quad P \in \mathbb{R}$ mine productivity

$\quad N \in \mathbb{R}^{n_u \times n_\ell \times n_m}$ number of trucks in each loading–dumping cycle

1:  $P \leftarrow 0$

2:  $N_{i_u, i_\ell, i_m} \leftarrow 0, \; \forall i_u, i_\ell, i_m$

3:  $t_{c, i_u, i_\ell, i_m} \leftarrow 2 \frac{d_{i_u, i_\ell}}{v_{i_m}} + t_{u, i_u, i_m} + t_{\ell, i_\ell, i_m}, \; \forall i_u, i_\ell, i_m$ $\qquad\qquad\qquad$ ▷ cycle time

4:  $P_c \leftarrow \frac{L_{i_m}}{t_{c, i_u, i_\ell, i_m}}, \; \forall i_u, i_\ell, i_m$ $\qquad\qquad\qquad\qquad\qquad\qquad$ ▷ cycle productivity

5:  $n_{r, i_m} \leftarrow n_{t, i_m}, \; \forall i_m$ $\qquad\qquad\qquad\qquad\qquad\qquad\qquad\qquad$ ▷ remaining trucks

6:  $a_{u, i_u} \leftarrow 0, \; \forall i_u$ $\qquad\qquad\qquad\qquad\qquad\qquad\qquad\qquad$ ▷ dumping site occupation

7:  $a_{\ell, i_\ell} \leftarrow 0, \; \forall i_\ell$ $\qquad\qquad\qquad\qquad\qquad\qquad\qquad\qquad$ ▷ loading site occupation

8:  $b_{i_u, i_\ell, i_m} \leftarrow 1, \; \forall i_u, i_\ell, i_m$ $\qquad\qquad\qquad\qquad\qquad\qquad\qquad$ ▷ remaining cycles

9:  **while** any $n_{r, i_m} > 0$ and any $b_{i_u, i_\ell, i_m} = 1$ **do**

10:  $\quad (i_u^\star, i_\ell^\star, i_m^\star) \leftarrow \arg\max_{i_u, i_\ell, i_m} P_{c, i_u, i_\ell, i_m} : b_{i_u, i_\ell, i_m} = 1$ $\qquad$ ▷ best cycle

11:  $\quad n_{\min} \leftarrow \min\left\{ n_{r, i_m^\star}, \frac{(1 - a_{u, i_u^\star}) t_{c, i_u^\star, i_\ell^\star, i_m^\star}}{t_{u, i_u^\star, i_m^\star}}, \frac{(1 - a_{\ell, i_\ell^\star}) t_{c, i_u^\star, i_\ell^\star, i_m^\star}}{t_{\ell, i_\ell^\star, i_m^\star}} \right\}$ $\quad$ ▷ allocation

12:  $\quad a_{u, i_u^\star} \leftarrow a_{u, i_u^\star} + \frac{n_{\min} t_{u, i_u^\star, i_m^\star}}{t_{c, i_u^\star, i_\ell^\star, i_m^\star}}$

13:  $\quad a_{\ell, i_\ell^\star} \leftarrow a_{\ell, i_\ell^\star} + \frac{n_{\min} t_{\ell, i_\ell^\star, i_m^\star}}{t_{c, i_u^\star, i_\ell^\star, i_m^\star}}$

14:  $\quad n_{r, i_m^\star} \leftarrow n_{r, i_m^\star} - n_{\min}$

15:  $\quad P \leftarrow P + \frac{n_{\min} L_{i_m^\star}}{t_{c, i_u^\star, i_\ell^\star, i_m^\star}}$

16:  $\quad N_{i_u^\star, i_\ell^\star, i_m^\star} \leftarrow n_{\min}$

17:  $\quad b_{i_u^\star, i_\ell^\star, i_m^\star} \leftarrow 0$

18:  $\quad$ **if** $a_{u, i_u^\star} \geq 0.999$ **then**

19:  $\quad\quad b_{i_u^\star, i_\ell, i_m} \leftarrow 0, \; \forall i_\ell, i_m$

20:  $\quad$ **end if**

21:  $\quad$ **if** $a_{\ell, i_\ell^\star} \geq 0.999$ **then**

22:  $\quad\quad b_{i_u, i_\ell^\star, i_m} \leftarrow 0, \; \forall i_u, i_m$

23:  $\quad$ **end if**

24:  $\quad$ **if** $n_{r, i_m^\star} \leq 0.001$ **then**

25:  $\quad\quad b_{i_u, i_\ell, i_m^\star} \leftarrow 0, \; \forall i_u, i_\ell$

26:  $\quad$ **end if**

27:  **end while**

28:  **return** $P, N$

---

### 2.2. Simulation

The truck cycles can be simulated using a discrete event system simulation considering a dispatch rule, as depicted in Figure 2. Initially, one event is scheduled in the event calendar for each truck. In order to improve the simulation warm up, the trucks are distributed on dumping sites according to their respective cycle capacities obtained from the optimization model (5)–(9). The simulator then basically fires the next event in the calendar at each time, until the time horizon is reached. Each fired event schedules a new event in the calendar and changes the respective truck state. The scheduled times may follow a particular probability distribution, leading to a stochastic simulation and considering that uncertainty sources in different locations in the mine are unrelated, so that random numbers following

a probability distribution may be considered realizations of the respective uncertainty. Despite being simple, this computational model may be extended to consider further model details, as needed, and it is specially useful for short-term operation.

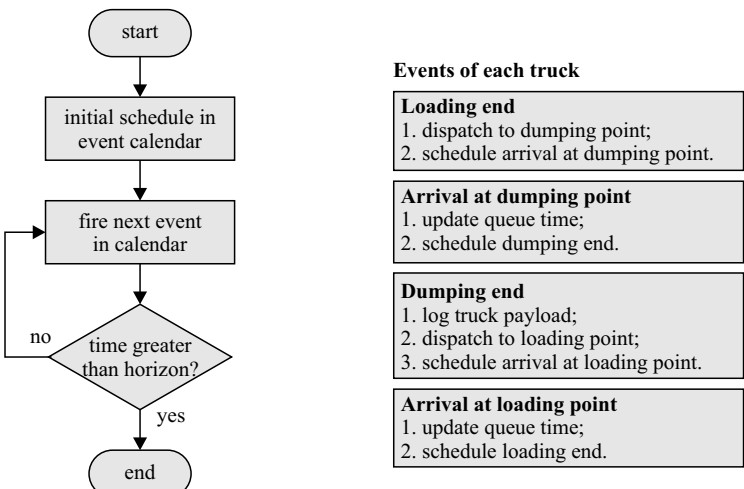

**Figure 2.** Discrete event simulator for mining truck cycles in a time horizon.

Dispatch Rule

Considering that the solution to the productivity optimization problem tends to be greedy, as considered by its respective approximate solution by Algorithm 1, it is consistent to also derive a greedy solution for the dispatch rule, where the fastest cycles are filled first. Hence, the proposed dispatch rule is to route trucks to services (i.e., loading or dumping) which will finish first in a prediction at the moment of dispatch, as depicted in Algorithm 2 to dispatch to a loading site. The algorithm to dispatch to a dumping site is analogous. This dispatch rule is easy to implement and, as shown in the results section of this paper, it leads to an almost optimal solution when compared to the productivity upper bounds provided by the optimization model (5)–(9),

---

**Algorithm 2** Dispatch rule to a loading site.

---

**Input**

$$
\begin{aligned}
t \in \mathbb{R} &\quad \text{current time} \\
d \in \mathbb{R}^{n_u \times n_\ell} &\quad \text{distance between the dumping and loading sites} \\
t_\ell \in \mathbb{R}^{n_u \times n_m} &\quad \text{truck loading time} \\
v \in \mathbb{R}^{n_m} &\quad \text{truck speed} \\
i_u \in \mathbb{N} &\quad \text{current dumping site} \\
i_m \in \mathbb{N} &\quad \text{truck model} \\
\tilde{t}_q \in \mathbb{R}^{n_\ell} &\quad \text{loading queue time}
\end{aligned}
$$

**Output**

$$
\begin{aligned}
i_\ell \in \mathbb{N} &\quad \text{next loading site} \\
\tilde{t}_q \in \mathbb{R}^{n_\ell} &\quad \text{loading queue time}
\end{aligned}
$$

1: $t_{h,j_\ell} \leftarrow d_{i_u,j_\ell}/v_{i_m}, \forall j_\ell$     ▷ transit time
2: $\tilde{t}_{j_\ell} \leftarrow \max\{\tilde{t}_{q,j_\ell}, t + t_{h,j_\ell}\}. \forall j_\ell$     ▷ loading starting time prediction
3: $i_\ell \leftarrow \arg\min_{j_\ell} \tilde{t}_{j_\ell} + t_{\ell,j_\ell}$     ▷ select the earliest loading finishing time prediction
4: $\tilde{t}_{q,i_\ell} \leftarrow \tilde{t}_{i_\ell} + t_{\ell,i_\ell}$     ▷ update queue time
5: **return** $i_\ell, \tilde{t}_q$

---

Following Algorithm 2, the predicted service (i.e., dumping and loading) finish time is tracked in a variable $\tilde{t}_q$. During the dispatch, the predicted service finish time $\tilde{t}_{j_s} + \tilde{t}_{s,j_s}$, $s \in \{u, \ell\}$, is calculated for each destination server $j_s$, considering the service time $t_{s,j_s}$ and the predicted service starting time $\tilde{t}_{j_s}$, which in turn depends on the transit time $t_{h,j_s}$, the

current time $t$, and the predicted last finish time of the respective server $\tilde{t}_{q,j_s}$. The truck is then dispatched to the server $i_s$ with the earliest predicted service finish time, which is set to the respective service finish time in $\tilde{t}_{q,is}$. For stochastic simulations, the mean values of the respective probability distributions are considered in the prediction.

## 3. Results

The main objective of the tests in the next subsections is to show how tight the proposed upper bound for productivity can be, and this is verified by a physical realization obtained from a simulation using the simple proposed truck dispatch rule.

### 3.1. Simulated Productivity Convergence and Its Upper Bound

Figure 3 shows the productivity convergence for a discrete event system simulation for

$$
d = \begin{bmatrix}
4095 & 3427 & 2700 & 5200 & 5394 & 5189 & 4617 & 4615 & 2293 & 2617 & 2816 & 3135 & 3242 & 3587 & 3862 \\
4198 & 3029 & 2770 & 5300 & 5400 & 6626 & 4800 & 4800 & 3325 & 3516 & 3374 & 3436 & 3210 & 2638 & 4000 \\
3554 & 2907 & 2704 & 5509 & 5736 & 5534 & 4985 & 4995 & 1607 & 1932 & 2212 & 2475 & 2529 & 2926 & 4248
\end{bmatrix} \text{m}, \quad (10)
$$

$n_u = 3$ dumping sites, $n_\ell = 15$ loading sites, $n_m = 2$ truck models, dumping time $t_u$, given in Table 1, loading time $t_\ell$ in Table 2, truck speed $v$, given in Table 3, truck load $L$, given in Table 4, and a fully available truck number $n_t = (12, 9)$. These data were obtained from 7 months of real operation in the Pico mine in Brazil, where the truck models were, respectively, CAT-785C and CAT-789D. Pico is a mine in Itabirito city, in the state of Minas Gerais in Brazil, which has been in operation since 1942 with a processed iron ore production capacity of 22Mt/year. Notice from Figure 3 that one day of simulation is enough for a long-term productivity estimation (actually, the productivity looks steady after 12 hours of model simulation), and also that the simulated productivity is never larger than the upper bound. The gap between the simulated steady state productivity and its upper bound is mostly explained by the heterogeneous fleet, in particular, due to the 25% difference in the load capacity between the two truck models. A queue priority for more productive models would make the gap smaller, but this is beyond the scope of this paper and could be a research theme for further investigation. The next section shows a homogeneous fleet simulation, which leads to much tighter gaps.

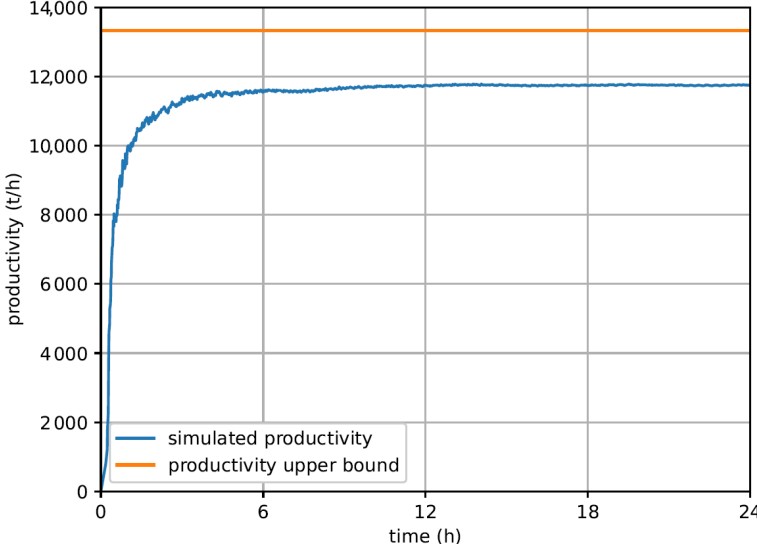

**Figure 3.** Productive convergence using a discrete event system simulation for the mine, which must be bellow the upper bound given by the linear optimization problem (5)–(9).

**Table 1.** Dumping time $t_u$ for each truck model (i.e., they are the same for every dumping site) following a triangular distribution in the interval $[a, b]$ with the mode at $c$.

| Truck Model $i_m$ | Minimum $a$ (s) | Maximum $b$ (s) | Mode $c$ (s) |
|---|---|---|---|
| 1 | 23 | 47 | 35 |
| 2 | 30 | 54 | 42 |

**Table 2.** Loading time $t_\ell$ for each truck model (i.e., they are the same for every loading site) following a triangular distribution in the interval $[a, b]$ with the mode at $c$.

| Truck Model $i_m$ | Minimum $a$ (s) | Maximum $b$ (s) | Mode $c$ (s) |
|---|---|---|---|
| 1 | 146 | 298 | 222 |
| 2 | 185 | 349 | 267 |

**Table 3.** Truck speed $v$ for each truck model following a triangular distribution in the interval $[a, b]$ with the mode at $c$.

| Truck Model $i_m$ | Minimum $a$ (km/h) | Maximum $b$ (km/h) | Mode $c$ (km/h) |
|---|---|---|---|
| 1 | 15 | 31 | 23 |
| 2 | 17 | 33 | 25 |

**Table 4.** Truck load $L$ for each truck model following a triangular distribution in the interval $[a, b]$ with the mode at $c$.

| Truck Model $i_m$ | Minimum $a$ (t) | Maximum $b$ (t) | Mode $c$ (t) |
|---|---|---|---|
| 1 | 138 | 148 | 143 |
| 2 | 189 | 201 | 195 |

*3.2. Fleet Sizing*

Figure 4 shows the increase in mine productivity with the number of available CAT-789D trucks until a saturation point is reached, where more trucks will only generate queues. This case study considers the distance (10) between $n_u = 3$ dumping sites and $n_\ell = 15$ loading sites, as well as the parameters in Tables 1–4 for truck model 2 (i.e., CAT-789D). Notice that the greedy search solution matches the optimal solution, which is typical, but does not always happen (sometimes the greedy search productivity is a little smaller just before the saturation point). This quantifies the optimality of the greedy search. Notice from Figures 4 and 5 that the simulated productivity, which is a feasible realization of trucks in the mine, starts close to the optimal productivity upper bound given by the solution of (5)–(9), gradually becoming smaller than the upper bound because of queues as the number of trucks increases. The simulated productivity ends close to the upper bound again (with a gap inferior to 2%), as the number of trucks is large enough to occupy all loading and dumping services without gaps. This quantifies the optimality of the dispatch rule and also quantifies how tight the upper bound can be. Finally, notice also that greater uncertainty in the simulation leads to lower productivity due to unforeseen queues, whose gap becomes smaller as the number of trucks gets further from the saturation point. For 50% of the uncertainty (i.e., triangular distribution with mean $c = \mu$, minimum $a = 0.5\mu$ and maximum $b = 1.5\mu$), the gap between the simulated productivity and its upper bound can become as large as 25%. Finally, notice that as the number of trucks gets large enough, all simulated productivity tends toward the upper bound (curiously, all with a relative error about 2%), no matter what the uncertainty involved is and so does the greedy search solution. This result shows how robust to uncertainty the proposed dispatch rule is, as well as how optimal the greedy search solution is. These results are depicted in detail and obtained for more mine instances in the next section.

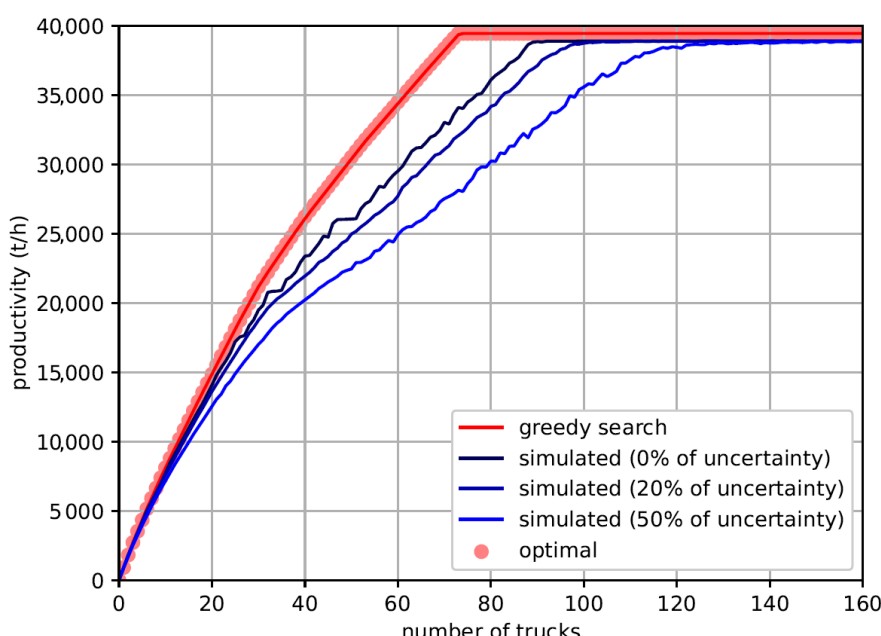

**Figure 4.** Mine productivity as a function of number of trucks available in the mine. Models with uncertainty $p \in \{20\%, 50\%\}$ consider the average productivity of 30 runs using symmetric triangular distributions with the mean at $\mu$, whose minimum and maximum values are at $(1 - p)\mu$ and $(1 + p)\mu$, respectively.

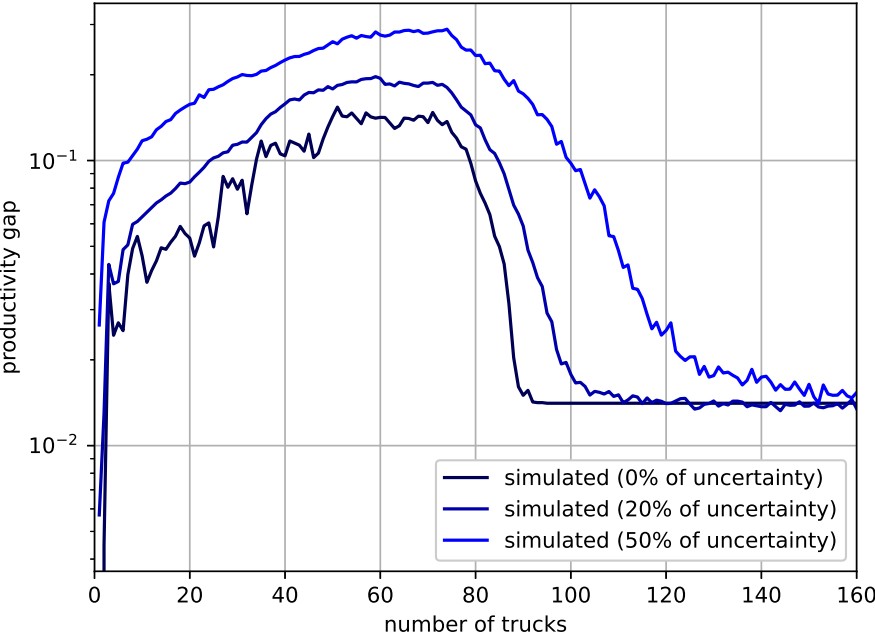

**Figure 5.** The mine's realized productivity gap to the upper bound as a function of the number of trucks available in the mine. Models with uncertainty $p \in \{20\%, 50\%\}$ consider the average productivity of 30 runs using symmetric triangular distributions with the mean at $\mu$, whose minimum and maximum values are at $(1 - p)\mu$ and $(1 + p)\mu$ respectively.

### 3.3. A Tight Upper Bound

Figure 6 shows how small the gap between the productivity upper bound and its realization is with and without uncertainty. Figure 6 also shows the greedy search solution gap to the exact linear programming solution. Notice how optimal the greedy search approximation tends to be. Indeed, it is very often optimal up to the machine floating

point precision of $2^{-52} \approx 10^{-16}$. Furthermore, the gap between the upper bound and its realization with the simulation is relatively small (typically 1%) for a enough large number of trucks, even for greater uncertainties. The independence on the uncertainty is due to the greater number of trucks that fill in the time gaps caused by prediction errors of the dispatch rule. This experiment considers the average gap for ten different instances with three dumping points, up to fifteen loading points and one hundred and sixty trucks (a relatively large number of trucks, as shown in Figure 4 for the largest instance), where the distances between the loading and dumping points are random numbers with equal probabilities in the range $[500, 5000]$ m, the loading times are random numbers with equal probabilities in the range $[240, 300]$ s, and the unloading times are random numbers with equal probabilities in the range $[50, 70]$s.

It is also noteworthy in Figure 6 that the gap to the upper bound does not vary much over the 15 instance sizes simulated on 10 different mine parameters (i.e., 150 different mine instances). This empirical evidence may be investigated further in order to find any justifying reason, which could be used to refine the dispatch rule or even the upper bound. Finally, the simulated mine productivity (with or without uncertainty) is consistently below the upper bound, which validates the upper bound, but it is also noteworthy that the productivity asymptotic convergence from below (as shown in Figure 3) is fundamental for this numerical result.

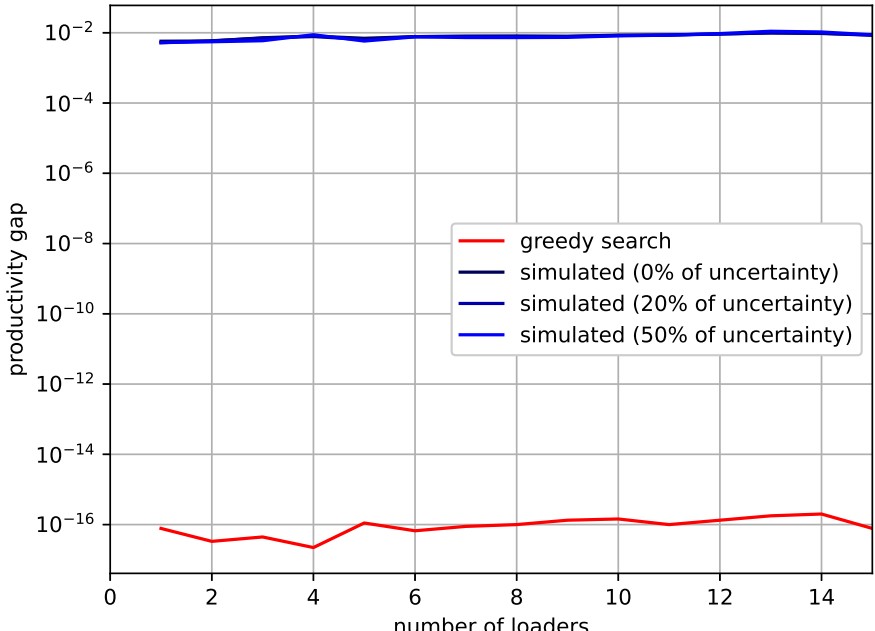

**Figure 6.** Average mine realized productivity gap to the upper bound for 10 different random instances with 160 trucks as a function of the number of loaders available in the mine. Models with uncertainty $p \in \{20\%, 50\%\}$ consider the average productivity of 30 runs using symmetric triangular distributions with the mean at $\mu$, whose minimum and maximum values are at $(1 - p)\mu$ and $(1 + p)\mu$ respectively.

## 4. Conclusions

The mine productivity's upper bound proposed in this paper is considerably tight when the number of trucks becomes large and it can be quickly obtained, especially if the proposed greedy search is used. This makes it suitable for the long-term planning of a mine (e.g., truck and shovel allocation problem). On the other hand, if the simulation is slower, it provides a physical realization of the mine operation. Furthermore, more details can be added to the simulation, making it suitable for short-term planning (e.g., truck dispatch problem).

The simple dispatch rule proposed in this paper leads to a productivity level close the its upper bound for homogeneous fleets, which indicates the suitability of the rules. When queues in the mine are small or when there are too many trucks in the mine, the proposed dispatch rule tends to provide a productivity level close to its upper bound. Considering that the dispatch rule is greedy, the good agreement between the greedy search and the linear programming productivity upper bound supports the good performance of the dispatch rule.

The approach introduced in this paper to couple long-term and short-term policies may serve to derive new objectives other than maximization of productivity (e.g., the maximization of shovel or truck utilization or the maximization of adherence to production specifications, as proposed in other papers) and their respective dispatch rules. Furthermore, as an instance of this approach, this paper introduces a new simple-to-implement and fast-to-run upper bound and its respective dispatch rule to maximize mine production from long-term planning to short-term operation with an agreement between the long-term planning and short-term operation of as close as 2%.

**Author Contributions:** A.C.L. has conceptualized the paper and conducted the experiments. F.L.B.C. has provided the data and analyzed the results. P.V.A.B.d.V. has validated, reviewed and edited the text. All authors have read and agreed to the published version of the manuscript.

**Funding:** This work was funded by CNPq under grant 304506/2020-6.

**Data Availability Statement:** This paper contains all data necessary for reproducing it.

**Acknowledgments:** This work was supported by CNPq, CAPES, and FAPEMIG, Brazil.

**Conflicts of Interest:** Authors Adriano Chaves Lisboa and Pedro Vinícius Almeida Borges de Venâncio were employed by the company Gaia, Solutions on Demand, Belo Horizonte, Brazil. Author Felipe Luz Barbosa Castro was employed by the company Vale S.A., Nova Lima, Brasil. The authors declare that the research was conducted in the absence of any commercial or financial relationships that could be construed as a potential conflict of interest.

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
