# Peer review of "Mine Productivity Upper Bounds and Truck Dispatch Rules"

_mining, doi:10.3390/mining3040043_

Round 1

Reviewer 1 Report

Comments and Suggestions for Authors

The comments are attached.

Comments on the Quality of English Language

Minor editing of English language required

Author Response

Thank you very much for your comments. We have carefully considered each one of them in the manuscript, without restrictions.

Reviewer 2 Report

Comments and Suggestions for Authors

The papers presents and approach using linear programming in order to determine the upper bound of and open pit mine productivity. It is based on the trucks load/dump cycles . This is applied/verified in a case study in an open pit mine in Brazil showing that the simulation results are applicable into real world conditions.

In my oppinion the paper is within the topic of the journal.

It is well documented, the theoretical aspects are also scientifically sound.

Introduction could be a bit more developed with what other scholars studied in this field, and to highlight what is different/better in the authors' approach as compared to literature in the field. Some more citation  (recent and from strong journals) could add value to the paper.

Also in the discussion/results/conclusion part it should be emphasised what is the advantage of the current presented method compared to other approaches. Why is it better or why it should be used in real world in comparison to existing methods?

Other that that, the paper is well written and also formatted according to MDPI rules, so if the small ammendents above are solve it can be published.

Additional comments

   1. What is the main question addressed by the research?

A question to ask the authors in my opinion. As far as i understand, it is a paper that aims to improve / maximise open mine production using a new linear programming tool (based on Greedy algorithms) with a case study in a Brasilian Mine.

    2. Do you consider the topic original or relevant in the field? Does it
    address a specific gap in the field?

The topic of open pit mining transportation and its optimisation is very vast in order to be able to produce 100% original or very new content. However I feel that is is relevant to the topic of a journal "Mining". It addresses the productivity of open pit mining in regard to its transport and haulage operation (ie truck load/dump cycles).

    3. What does it add to the subject area compared with other published material?

It is mandatory to present the research in comparison to similar topics found in recent literature, and highlight the strenghts and novelty their study. Weaknesses and future research direction would also increase the value of the paper.

    4. What specific improvements should the authors consider regarding the methodology? What further controls should be considered?

Not really. Methodology and theoretical part is good.

    5. Are the conclusions consistent with the evidence and arguments presented and do they address the main question posed?

Somewhat yes. Conclusion should prove that the approach is valid, and that the obtained results are better than those obtained by previous scholars / methods.  Highlight why this method is better / why this should be used in real world in comparison to other methods?

    6. Are the references appropriate?

I suggested in the review report that more references should be cited, ideally published in strong journals within the last years (ie recent ones)

    7. Please include any additional comments on the tables and figures.

Tables and figures are good quality.

Regards

Reviewer 3 Report

Comments and Suggestions for Authors

1. It is recommended to add detailed data to the abstract to directly express the effectiveness of the algorithm in improving productivity.

2. In Section 2, Figure 1, it is suggested to explain why the model does not take into account the queue time of trucks. Many studies have proven that truck queue times have a significant impact on the efficiency of heterogeneous and homogeneous fleets.

3. In the summary of 2.1, it is suggested to elaborate on the relationship between the upper bound of mine productivity and the truck scheduling rules, and elaborate on how the two affect each other in the conclusion.

4. In the summary of 2.1.1, it is suggested to improve the greedy search algorithm to increase originality.

5. In the summary, 2.2 Whether the diversity of the sample is taken into account.

6. In the summary, is it convincing to show only one result?

7. In the summary of 3.2, do you need to consider more factors, such as the influence of heavy load transportation distance, empty transportation distance, waiting time, slope and other factors?

8. Please describe in detail the site of the open-pit mine in Brazil.

Comments on the Quality of English Language

Accurate language expression.
